# ATR Contributes More Than ATM in Intra-S-Phase Checkpoint Activation after IR, and DNA-PKcs Facilitates Recovery: Evidence for Modular Integration of ATM/ATR/DNA-PKcs Functions

**DOI:** 10.3390/ijms23147506

**Published:** 2022-07-06

**Authors:** Aashish Soni, Xiaolu Duan, Martin Stuschke, George Iliakis

**Affiliations:** 1Division of Experimental Radiation Biology, Department of Radiation Therapy, University Hospital Essen, University of Duisburg-Essen, 45147 Essen, Germany; aashish.soni@uk-essen.de (A.S.); martin.stuschke@uk-essen.de (M.S.); 2Institute of Medical Radiation Biology, University Hospital Essen, University of Duisburg-Essen, 45147 Essen, Germany; xiaolu.duan@uk-essen.de; 3German Cancer Consortium (DKTK), Partner Site University Hospital Essen, and German Cancer Research Center (DKFZ), 45147 Essen, Germany

**Keywords:** DNA Double Strand Breaks (DSB), DNA-PK, ATM, ATR, intra-S-phase checkpoint, ionizing radiation (IR), DSB end resection (resection)

## Abstract

The intra-S-phase checkpoint was among the first reported cell cycle checkpoints in mammalian cells. It transiently slows down the rate of DNA replication after DNA damage to facilitate repair and thus prevents genomic instability. The ionizing radiation (IR)-induced intra-S-phase checkpoint in mammalian cells is thought to be mainly dependent upon the kinase activity of ATM. Defects in the intra-S-phase checkpoint result in radio-resistant DNA synthesis (RDS), which promotes genomic instability. ATM belongs to the PI3K kinase family along with ATR and DNA-PKcs. ATR has been shown to be the key kinase for intra-S-phase checkpoint signaling in yeast and has also been implicated in this checkpoint in higher eukaryotes. Recently, contributions of DNA-PKcs to IR-induced G_2_-checkpoint could also be established. Whether and how ATR and DNA-PKcs are involved in the IR-induced intra-S-phase checkpoint in mammalian cells is incompletely characterized. Here, we investigated the contributions of ATM, ATR, and DNA-PKcs to intra-S-phase checkpoint activation after exposure to IR of human and hamster cells. The results suggest that the activities of both ATM and ATR are essential for efficient intra-S-phase checkpoint activation. Indeed, in a wild-type genetic background, ATR inhibition generates stronger checkpoint defects than ATM inhibition. Similar to G2 checkpoint, DNA-PKcs contributes to the recovery from the intra-S-phase checkpoint. DNA-PKcs–deficient cells show persistent, mainly ATR-dependent intra-S-phase checkpoints. A correlation between the degree of DSB end resection and the strength of the intra-S-phase checkpoint is observed, which again compares well to the G2 checkpoint response. We conclude that the organization of the intra-S-phase checkpoint has a similar mechanistic organization to that of the G_2_ checkpoint in cells irradiated in the G_2_ phase.

## 1. Introduction

Cell cycle checkpoints are the caretakers of genomic stability [1,2,3]. They stop cell cycle progression following DNA damage, transiently or permanently, and thus provide time for repair [2,4,5] before the cell progresses further into the cell cycle; alternatively, checkpoints drive cells into programmed cell death or senescence. Failure to activate checkpoints leads to genomic instability through associated failures in replication or repair [6]. Replication-dependent radiosensitization is among the most promising approaches towards improving cancer treatment [7] and makes the mechanistic underpinnings of the S-phase checkpoint a relevant target.

Exposure to ionizing radiation (IR) triggers cell cycle checkpoints in all phases (G_1_, S, G_2_) of the cell cycle, except mitosis, mainly owing to the induction of DNA double-strand breaks (DSBs), one of the most deleterious DNA lesions [8,9]. Homologous recombination (HR) and classical non-homologous end joining (c-NHEJ) repair DSBs in mammalian cells and are essential for maintaining genomic stability. Alternative end-joining (alt-EJ) and single-strand annealing (SSA) are highly mutagenic repair pathways, which engage mainly to rescue failures of c-NHEJ and/or HR [10,11,12]. The G_1_ checkpoint prevents entry into the S phase and the G_2_ checkpoint prevents entry into mitosis of cells with damaged DNA. In addition, the intra-S-phase checkpoint ensures replication fidelity by transiently stopping cells from replicating damaged DNA, mainly by inhibiting the firing of new origins of DNA replication [13]. Defects of the intra-S-phase checkpoint give rise to radio-resistant DNA synthesis (RDS) following exposure to IR and have been shown to be a hallmark of carcinogenesis.

Phosphoinositide 3 kinases related protein kinases (PI3KK), ataxia telangiectasia mutated (ATM), ATM and RAD3-related (ATR), and DNA-dependent protein kinase catalytic subunit (DNA-PKcs) form the central core of kinases in DNA damage signaling following exposure to IR. According to the classical view, ATM is the master regulator of IR-induced checkpoints [14,15,16]. ATR is considered to function in an ATM-dependent manner recruited on RPA-coated single-stranded DNA generated after end resection at DSBs [15,17,18,19,20] (hereafter referred to simply as resection). DNA-PKcs guides DSB repair through classical non-homologous end joining (c-NHEJ) [16,21,22] and is considered to have no impact on cell cycle checkpoints [23,24,25].

We recently reported evidence that allowed us to modify and extend this classical view of G_2_ checkpoint signaling. In this revised view of checkpoint signaling, ATM, ATR, and DNA-PKcs are thought to form a module that differentially regulates the G2 checkpoint in cells irradiated in G_2_ phase versus S phase. Notably, this form of checkpoint regulation changes profoundly between low (in the range of 2 Gy or less) and high IR doses (in the range of 10 Gy or more) [26,27,28].

In the present report, we extend this form of analysis to the regulation of the intra-S-phase checkpoint. Specifically, we investigate the mutual interactions and dependencies among ATM, ATR, and DNA-PKcs in the regulation of this checkpoint. Since available assays for intra-S-phase checkpoint analysis lack sensitivity for measurements at low IR doses (less than 5 Gy), we restrict our analysis to high IR doses (≥10 Gy).

Notably, at high IR doses in the G_2_ phase, both ATM and ATR contribute to the regulation of the G_2_ checkpoint, and as a result, combined inhibition of both kinases is essential to abrogate the G_2_ checkpoint completely [26,27]. A similar correlation between ATM and ATR has also been found in the regulation of the G_1_ checkpoint at high IR doses [29]. On the other hand, irradiation in the S phase leads to the development of a solely ATR-dependent G_2_ checkpoint, with ATM now contributing to checkpoint recovery [27]. DNA-PK is crucial for the recovery from the G_2_ checkpoint for both irradiated S and G_2_ phase cells [26,27]. The potency of the IR-induced G_1_ or G_2_ checkpoint was also found directly proportional to the extent of DSB end resection [4,26,27,29], which further emphasizes its dependence on ATR.

ATM-deficient cells show clear defects in the activation of the IR-induced intra-S-phase checkpoint, which causes RDS by inhibiting CHK1- and CHK2-mediated proteolytic degradation of Cdc25A [30,31]. Although ATM and ATR (in an ATM-dependent manner) have been implicated in the regulation of intra-S-phase checkpoint signaling, their relative contributions and interplay remain incompletely characterized. While ATM is activated by IR-induced DSBs, ATR is activated in response to various genotoxins, such as hydroxyurea, UV rays, aphidicolin, etc. These differences in response occasionally hamper analysis of their mutual interactions. In the current study, we sought to investigate the role of ATR [28] and most importantly the interplay among ATM, ATR, and DNA-PKcs in the IR-induced intra-S-phase checkpoint.

We report that in contrast to the G2 checkpoint response of cells irradiated in the S phase, which is solely dependent on ATR, the intra-S-phase checkpoint, i.e., the checkpoint activated in S-phase cells that have also been irradiated in the S phase, is dependent on the activities of both ATM and ATR. This response is similar to the G_2_ checkpoint–activated in cells irradiated in the G_2_ phase. Similar to the IR-induced G_2_ checkpoint, DNA-PKcs is crucial for the recovery from the intra-S-phase checkpoint, which is also in agreement with our previous findings [32]. Interestingly, ATR dominates over ATM in signaling persistent intra-S-phase checkpoint in DNA-PKcs deficient or inhibited cells. Similar to the IR-induced G_2_ checkpoint, the magnitude of the intra-S-phase checkpoint correlates with the degree of resection. Inhibition of ATM or ATR individually, significantly reduces resection, and combined inhibition abolishes resection. Ablation, but not chemical inhibition, of DNA-PKcs exerts a significant increase in resection upon irradiation, and this feeds directly into the potentiation of the intra-S-phase checkpoint.

## 2. Results

### 2.1. The Activation of the Intra-S-Phase Checkpoint Is Strongly Dependent on the Functions of Both ATM and ATR

The impact of ATM and ATR on intra-S-phase checkpoint was first measured in wild- type Chinese hamster ovarian (CHO) cells. CHO cells offer an attractive model for DNA damage response (DDR) studies owing to the availability of relevant DSB repair mutants, which are employed in this study and are described later. Exponentially growing CHO cells were exposed to 10 Gy of IR in the presence or absence of ATM inhibitor (ATMi), and/or ATR inhibitor (ATRi), and DNA synthesis was measured at the indicated times after IR, using tritiated thymidine (^3^H) incorporation into DNA. The reduction in ^3^H incorporation reflects inhibition of DNA synthesis owing to the activation of the intra-S-phase checkpoint.

CHO cells show, at 1 h post-IR, about 40% reduction in DNA synthesis. DNA synthesis recovers eventually and reaches ~80% of controls within 3 h post-IR (Figure 1A). ATM inhibition partially abrogates the checkpoint that now causes only a 20% reduction in DNA synthesis, which remains relatively unchanged until 3 h (Figure 1B). ATR inhibition strongly abrogates the checkpoint, with only ~10% transient reduction in DNA synthesis at 0.5 h and a quick recovery to control levels at 1 h.

Indeed, an overshooting in DNA synthesis is noted at 3 h post-IR (Figure 1B). The combined inhibition of ATM and ATR results in nearly complete abrogation of the intra-S-phase checkpoint at all three levels: initiation, maintenance, and recovery (Figure 1C). Since ATR and its downstream target Chk1 have been shown to affect origin firing during unperturbed DNA replication [33,34,35,36,37], we additionally checked the impact of ATRi on DNA synthesis in unirradiated cells. A slight increase in DNA synthesis is observed at 1 and 3 h after ATRi treatment, when compared to 0.5 h (data not shown). This increase might explain the overshooting observed in DNA synthesis at 3 h after IR.

In addition to CHO cells, we investigated the impact of ATM and ATR on the intra- S-phase checkpoint in the human lung carcinoma cell line, A549. A549 cells show upon irradiation a robust checkpoint (~70% reduction in DNA synthesis) at 30 min post-IR (Figure 2A). The checkpoint remains strong until 1 h, with clear evidence of recovery at 3 h post-IR. Here again, ATM inhibition partially suppresses the checkpoint (Figure 2A), yielding only about a 40% reduction in DNA synthesis. ATR inhibition again exerts a stronger checkpoint abrogation (Figure 2B), resulting in only ~20% reduction in DNA synthesis. The combined inhibition of ATM and ATR abrogates the checkpoint to a similar extent (Figure 2C) as the inhibition of ATR alone. The above set of data, collectively, suggests that both ATM and ATR kinases contribute to the activation of the IR-induced intra-S-phase checkpoint in rodents as well as in human cells, with ATR exerting a slightly stronger effect than ATM (see below for a model of the mechanism).

### 2.2. DNA-PKcs Contributes to the Recovery from IR-Induced Intra-S-Phase Checkpoint

DNA-PKcs has been recently implicated in the recovery from IR-induced G_2_ checkpoint in cells irradiated either in the S or G_2_ phase of the cell cycle [26,27]. Earlier work provided evidence that DNA-PK is important for the recovery from IR-induced intra-S-phase checkpoint [32], and we sought to follow up on this, taking advantage of the availability of highly specific inhibitors of ATM, ATR, and DNA-PKcs. We first measured the effect of DNA-PKcs on intra-S-phase checkpoint using a specific small-molecule inhibitor (DNA-PKi) and DNA-PKcs mutants of human and CHO origin.

Inhibition of DNA-PK results in slightly stronger inhibition of DNA synthesis, as compared to untreated samples of CHO and A549 cells. Notably, DNA-PK–inhibited cells show persistent checkpoint, indicating suppression of checkpoint recovery in both rodent and human cells (Figure 3A,B). Similar responses could also be measured in DNA-PKcs mutants of human M059J, Chinese hamster (V3) origin, and HCT116 DNA-PK^−/−^ (Figure 3C,D and Appendix A respectively). These results implicate DNA-PKcs in intra-S-phase checkpoint recovery, as previously reported for the G_2_ checkpoint [26,27].

To investigate whether suppression of c-NHEJ-mediated DSB repair is solely responsible for the checkpoint potentiation observed in DNA-PK inhibited or mutant cells, we measured intra-S-phase checkpoint in the immortalized normal human fibroblast cells, 82-6 hTert, and compared the results to those obtained in two LIG4 mutants, 180BR and 411BR. We selected the LIG4 mutation for these experiments, as it is known to generate the strongest defect on DSB repair by c-NHEJ. The results summarized in Figure 3E show that 82-6 hTert, 180BR, and 411BR cells activate and recover from the checkpoint with similar kinetics, despite the expected longer persistence of DSBs in 180BR and 411BR cells.

We also tested, in similar experiments, HCT116 WT and their LIG4-deficient counterpart HCT116 LIG4^−/−^, and compared the results to those obtained with DNA-PK^−/−^ cells generated on the same genetic background (Appendix A). In this set of cell lines, activation of the intra-S-phase checkpoint is similar, but recovery deviates. While wild-type cells begin recovering DNA synthesis 2 h after IR, HCT116 LIG4^−/−^ cells begin recovering only after 3 h but do reach control levels at 6 h. On the other hand, their DNA-PKcs–deficient counterparts show sustained suppression of DNA replication at 6 h with no evidence of recovery.

XR1 cells, a CHO mutant in the LIG4 cofactor XRCC4, show activation of the intra-S-phase checkpoint, with kinetics similar to wild-type CHO cells and their DNA-PKcs–deficient counterparts, V3. However, while XR1 cells show delayed recovery, DNA-PKcs–deficient cells show sustained suppression of DNA replication (Figure 3D,F). We conclude that DNA-PK is essential for efficient recovery from the intra-S-phase checkpoint and that this effect is specific for DNA-PKcs and may not be entirely attributed to the associated inhibition of c-NHEJ that occurs to a greater extent in LIG4 mutants. It is evident that in some but not all cell lines, c-NHEJ suppression by defects in factors other than DNA-PKcs delays checkpoint recovery. The variability among cell lines of this effect and the underpinning parameters will require further study in future work. The question as to which kinase (ATM and/or ATR) sustains the persistent intra-S-phase checkpoint in a DNA-PKcs–deficient background is examined next.

### 2.3. Persistent Intra-S-Phase Checkpoint in DNA-PKcs Mutants Is Independent of ATM Activity

As shown above, ATM inhibition partly abrogates the intra-S-phase checkpoint in wt Chinese hamster and human cells (Figure 1A and Figure 2A, respectively). We tested whether ATM is responsible for the persistent intra-S-phase checkpoint in DNA-PKcs deficient cells. Human glioma wild-type (M059K) and DNA-PK mutant (M059J) cells, as well as a CHO mutant of DNA-PKcs, V3, were treated with ATMi, and the percentage of DNA synthesis was measured. Wt M059K cells show significant abrogation of the checkpoint (Figure 4A), which is similar to that observed in CHO and A549 cells (Figure 1A and Figure 2A, respectively). Notably, ATM inhibition has a much smaller effect on V3 (Figure 4C) and particularly on M059J (Figure 4B) cells, suggesting that ATM’s contribution to the intra-S-phase checkpoint diminishes in DNA-PKcs–deficient cells. Similar results are obtained with XR1 cells (Appendix A).

### 2.4. ATR Is the Key Activator of the Persistent Intra-S-Phase Checkpoint in DNA-PKcs Mutants

Using an experimental set up similar to that described above for ATMi, we analyzed the impact of ATR inhibition on an intra-S-phase checkpoint in DNA-PKcs mutant cells. Cells were treated with ATRi, and the percentage of DNA synthesis was measured in wt and DNA-PKcs–deficient mutants. Wild-type M059K cells (Figure 5A) show clear abrogation of the checkpoint, similar in magnitude to that observed in CHO or A549 cells (Figure 1B and Figure 2B, respectively). Notably, ATR inhibition also resulted in nearly complete abrogation of the intra-S-phase checkpoint in M059J and V3 cells (Figure 5B,C), which demonstrates that the strong and persistent intra-S-phase checkpoint in DNA-PKcs mutants is mainly sustained by ATR. ATRi also abrogates the checkpoint observed in XR1 cells (Appendix A).

### 2.5. The Degree of DNA End-Resection at DSBs in the S-Phase Correlates with the Strength of the Intra-S-Phase Checkpoint

Our recent studies on IR-induced checkpoint signaling have shown a correlation between the degree of resection and the activation of G_1_ or G_2_ phase checkpoints. Indeed, ATM-, ATR-, and DNA-PKcs–mediated modulation of resection is a key factor in these responses [4,26,27,29]. Here, we examined whether the degree of resection in S-phase cells correlates with the strength of IR-induced intra-S-phase checkpoint. We measured resection in CHO cells after exposure to 20 Gy of IR, in the presence or absence of the above-discussed inhibitors of these kinases. RPA70 signal and EdU signal were measured by FACS at 3 h post-IR to enable cell-cycle specific analysis—here analysis in S-phase cells.

RPA70 intensity in the S phase is measured by gating S-phase cells as shown in Figure 6A. Irradiated CHO cells show an increase in RPA70 signal when compared to 0 Gy controls, indicating IR-induced DNA end resection (Figure 6B upper panel). DNA-PKcs inhibition (Figure 6B second panel from top) has no effect on resection, as reported earlier [26,27,38]. Inhibition of ATM (Figure 6B third panel from top) or ATR individually (Figure 6B fourth panel from top) partly suppresses resection. Notably, combined inhibition of ATM and ATR results in nearly complete inhibition of resection (Figure 6B bottom panel). Suppression of resection through ATMi or ATRi individually, but particularly in combination, reflects well the observed impact of these kinases on the intra-S-phase checkpoint. In addition, the DNA-PKcs mutant, V3, shows clearly elevated resection in S phase when compared to wild-type CHO cells (Figure 6C), which is in line with similar results obtained in G_2_-phase irradiated cells [26,27,38].

## 3. Discussion

### 3.1. ATM and ATR Contribute to the Intra-S-Phase Checkpoint

Multiple studies have implicated ATM as the main kinase for the activation of IR-induced cell cycle checkpoints in mammalian cells [39,40], including the intra-S-phase checkpoint investigated here [14,41]. ATR, owing to the requirement for RPA-coated ssDNA for its activation [20], is considered to be mainly involved in the UV-induced intra-S-phase checkpoint and the regulation of DNA replication stress [14,16,42]. As a consequence, the contributions of ATR to IR-induced checkpoints are typically estimated as rather secondary and partly ATM-dependent [26,27].

However, recent results from our laboratory implicate ATR in IR-induced checkpoint responses much stronger than hitherto thought and suggest that ATM, ATR, and DNA-PKcs operate as a module to regulate IR-induced checkpoints. Notably, the putative ATM/ATR/DNA-PKcs module displays striking mechanistic adaptations that accommodate not only the progression of the irradiated cell through the cell cycle but also increases in IR dose, i.e., increases in the load to DSBs present in the genome [26,27,29]. In the current paper, we examined for the first time the crosstalk among ATM, ATR, and DNA-PKcs in the regulation of IR-induced intra-S-phase checkpoint. This study, as well as our previous studies [26,27], significantly benefited from the relatively recent availability of highly specific inhibitors of ATM, ATR, and DNA-PKcs, which dramatically ease experimentation, complement genetic approaches, and help to overcome problems associated with the lethality of genetic ATR ablation.

Our results show that, in line with previous studies [28], both ATM as well as ATR contribute to the activation of the intra-S-phase checkpoint and that indeed the contribution of ATR is stronger (Figure 7). The stronger effect of ATR inhibition in the suppression of the intra-S-phase checkpoint is notable because we previously showed that ATM remains functional in ATR inhibited cells [26]. Because combined application of ATMi and ATRi generates an effect only slightly stronger than that of ATRi alone, we surmise that in the absence of ATR, ATM is unable to significantly contribute to the activation of the intra-S-phase checkpoint. This raises ATR to the key regulator of the intra-S-phase checkpoint.

These results are fully in line with those recently published for G_2_-checkpoint activation in cells also irradiated in G_2_ phase [26]. Indeed, in this case, and at low doses of IR, ATM and ATR epistatically regulate the G2 checkpoint, and inhibition of either kinase abrogates the checkpoint almost completely. On the other hand, at high IR doses that approach the range of doses used in the present study, ATM and ATR act independently, and inhibition of both is required to fully suppress the checkpoint. This is similar to what we observe for the intra-S-phase checkpoint in some cell lines.

Strikingly, when cells are irradiated in S phase, as was the case in all experiments presented here, the G_2_ checkpoint activated when cells reach the G_2_ phase uses ATM and ATR in an entirely different manner. Indeed, under these conditions, checkpoint activation relies entirely on ATR, and ATM contributes to its recovery as does also DNA-PKcs (see below) [27]. A comparison of these results to those presented here uncovers highly interesting and potentially relevant adaptations in the crosstalk between ATM and ATR, as S-phase irradiated cells progress through the cell cycle and arrest in the following G_2_ phase.

Why do the roles of ATM and ATR change among the two types of checkpoints experienced by cells irradiated in S phase? One reason might be that in the period of the cell cycle where the intra-S-phase checkpoint is activated, the genome is present in both unreplicated G_1_-like and replicated G_2_-like forms. However, when S-phase cells enter the G_2_ phase, DNA is exclusively in G_2_-like form. Consequently, G_1_-like DNA in S phase requires both ATM and ATR for checkpoint activation, as reported recently for the G_1_ checkpoint at high IR doses [29]. However, the G_2_-like DNA activates the checkpoint using mainly ATR. It has been reported that the G_1_ and the intra-S-phase checkpoints involve a similar signaling cascade [43,44,45,46,47], and in accordance, the combined action of ATM and ATR at high DSB loads.

The intra-S-phase checkpoint is transient in nature, lasts only a few minutes, and may not provide enough time for the completion of DNA repair. Thus, cells with unrepaired DNA damage may enter the G_2_ phase and arrest at the G_2_/M border [48,49]. This checkpoint relies solely on ATR [27] and connects to HR but not c-NHEJ [4,50]. Indeed, S-phase–irradiated cells enter the G_2_ phase up to 8 h after IR, which leaves enough time for c-NHEJ to complete. It is likely, therefore, that the coordination of the G_2_ checkpoint with c-NHEJ is lost, leaving HR and thus ATR as the sole contributor. In other words, the contribution of ATM to checkpoint activation is an early event and can only be observed in the phase of the cell cycle, where cells are irradiated. When cells move to the subsequent phase of the cell cycle, ATM mainly contributes to checkpoint recovery. Whether the intra-S-phase checkpoint facilitates DSB repair and how, and which repair pathways are favored during its activation requires further investigations.

The observations summarized here on the activation in S-phase–irradiated cells of the intra-S-phase and the G_2_-checkpoint and the roles of ATM and ATR in these responses are schematically shown in Figure 7.

### 3.2. DNA-PKcs Defects Cause Mainly ATR-Dependent Potentiation of Intra-S-Phase Checkpoint

Following successful repair or following adaptation to DNA damage, cells eventually re-enter the cell cycle through a process known as checkpoint recovery. Efficient recovery from a checkpoint is essential for the completion of checkpoint responses. DNA-PKcs has been implicated in the recovery from IR-induced G_2_ and intra-S-phase checkpoint [26,27,32,51,52,53], suggesting c-NHEJ-independent functions for the kinase.

In the present study, we confirmed that DNA-PKcs is crucial for the recovery from IR-induced intra-S-phase checkpoint, as reported earlier by Guan J. et al. [32]. Inhibition of DNA-PKcs causes a persistent S-phase checkpoint, which mainly depends on ATR activity. This response is similar to that reported earlier for the recovery from the G_2_ checkpoint, both for cells irradiated in S, as well as in the G_2_ phase of the cell cycle [26,27]. In addition, DNA-PKcs mutant cells show increased resection in irradiated S-phase cells, which likely causes stronger ATR activation and potentiation of the intra-S-phase checkpoint. It has also been reported that ATR regulates origin firing upon replication stress and that DNA-PK along with Chk1 serve as a backup pathway in this regulation when ATR is inhibited, highlighting additional interactions between ATR and DNA-PKcs [54].

The possibility that the delayed intra-S-phase checkpoint recovery in DNA-PKcs–deficient cells is due to compromised DSB repair through a defect in c-NHEJ is unlikely, as *XRCC4* and *LIG4* mutants, despite the potentiated checkpoint, exhibited checkpoint recovery when compared to their corresponding DNA-PK deficient counterparts (Figure 3E,F and Appendix A). In addition, recent studies from our group show that only DNA-PKcs mutation causes hyperresection upon irradiation in G_2_-phase cells, while cells with defects in other c-NHEJ factors show normal resection levels, comparable to control cells [26,38]. It has been shown that DNA-PKcs may crosstalk with ATM and ATR [53,54,55,56,57], suggesting a non-canonical, c-NHEJ–independent, function. Our results also suggest a non-canonical DDR function of DNA-PKcs in the IR-induced intra-S-phase checkpoint (Figure 7).

The results presented in this paper uncover the potential for coordinated functions of ATM, ATR, and DNA-PKcs in the intra-S-phase checkpoint (Figure 7), as has been reported earlier for IR-induced G_1_ and G_2_ checkpoint responses at high DSB loads [4,26,27,29]. One limitation of the current study is the absence of low dose results, owing to the reduced sensitivity of the assay employed. As a consequence, we could not reliably measure the intra-S-phase checkpoint at doses below 10 Gy. Thus, the interplay among ATM, ATR, and DNA-PKcs kinases in the regulation of the intra-S-phase checkpoint at low DSB loads remains elusive. Which DSB repair pathway/s engage during the intra-S-phase checkpoint will also need further detailed investigations.

## 4. Methods and Materials

### 4.1. Cell Culture and Growth Conditions

All cells were grown at 37 °C in an atmosphere with 5% CO_2_ and 95% air. HCT116, wt, *LIG4*^−/−^, and DNA-PK^−/−^, Chinese hamster, CHO10B4, XR1, and V3 cells were maintained in McCoy’s 5A medium supplemented with 10% fetal bovine serum (FBS). A549 cells were maintained in McCoy’s 5A medium. Culture media were supplemented with 10% FBS. The DNA *LIG4* deficient, 180BRM, 411BR, and the normal human fibroblasts, 82-6 hTert, were maintained in minimum essential medium (MEM), supplemented with 10% FBS and 1% non-essential amino acids (NEA).

### 4.2. Chemicals and Inhibitors

Then, 2-morpholin-4-yl-6-thianthren-1-yl-pyran-4-one (KU55933, ATM inhibitor, depicted with ATMi, Calbiochem; IC50 ATM  =  13 nM; IC50 ATR  =  100 μM; IC50 DNA-PKcs  =  2.5 μM) was dissolved in DMSO (Sigma-Aldrich, St. Louis, MO, USA) at 10 mM and used at 10 μM final concentration. The 8-(4-Dibenzothienyl)-2-(4-morpholinyl)-4H-1-benzopyran-4-one (NU7441, DNA-PKcs inhibitor, termed here as DNA-PKi, Tocris Bioscience; IC50 DNA-PKcs  =  13 nM; IC50 ATM  =  100 μM; IC50 ATR  =  100 μM) was dissolved in DMSO at 10 mM and used at 5 μM final concentration. The 3-Amino-6-[4-(methylsulfonyl)phenyl]-N-phenyl-2-pyrazinecarboxamide (VE821, ATR inhibitor, termed here as ATRi, Haoyuan Chemexpress; IC50 ATR  =  26  nM; IC50 ATM > 8  μM; IC50 DNA-PKcs  =  4.4  μM) was dissolved in DMSO at 10  mM and used at 5 μM final concentration. All inhibitors were administered 1 h before irradiation and were maintained during the entire duration of the experiment.

### 4.3. Irradiation

Irradiations were carried out with an X-ray machine (GE-Healthcare, Chicago, IL, USA) operated at 320 kV, 10 mA with a 1.65 mm Al filter (effective photon energy approximately 90 kV), at a distance of 50 cm and a dose rate of approximately 1.3 Gy/min. The dosimetry was performed with a PTW and/or a chemical dosimeter, which was used to calibrate an in-field ionization monitor. Even exposure to radiation was ensured by rotating the radiation table. Cells were returned to the incubator immediately after IR.

### 4.4. Measurement of Radio-Resistant DNA Synthesis

DNA synthesis was measured by tritiated thymidine (^3^H) incorporation in exponentially growing cells. ^3^H was added for 15 min prior to collection of the indicated time point after IR exposure. Cells were trypsinized, counted, and loaded on glass microfiber filters (Whatman GF/A). The filters were washed 3X with a 10% solution of trichloroacetic acid (TCA), washed (3X) with deionized water, and incubated for 12–14 h in 0.5 mL of 0.5 N NaOH at 60 °C. The samples were subsequently neutralized with 0.5 mL of 0.5 N HCl, supplied with 10 mL scintillation fluid, and counted for ^3^H-activity in a liquid scintillation counter (Perkin Elmer Tri-Carb^®^ 2910TR, Waltham, MA, USA). The rate of DNA synthesis was calculated from these measurements and is presented as the percentage of values measured in sham-irradiated controls.

### 4.5. Flow Cytometry-Based DSB end Resection Assay

DNA end resection was measured using a RPA70 staining-based flow cytometric method [4,26,27,58]. Briefly, exponentially growing cells were pulse-labeled for 30 min with 5 µM 5-ethynyl-2′-deoxyuridin (EdU). EdU-containing media were removed, and cells were rinsed once with pre-warmed PBS, supplemented with fresh pre-warmed growth medium prior to exposure to X-rays. Cells were collected at 3 h after IR by trypsinization. Non-irradiated cells were collected at 1 h after the EdU pulse. Unbound RPA was extracted by treating cells with ice-cold phosphate-buffered saline (PBS) containing 0.2% Triton X-100 for 2 min on ice. Cells were spun-down and fixed for 15 min in PFA solution at room temperature (RT). Cells were then blocked with PBG buffer overnight at 4 °C until preparation for immunostaining. Cells were incubated with a monoclonal antibody raised against RPA70, which was kindly provided by Dr. Hurwitz [59] for 1.5 h at RT. Cells were washed twice with PBS and incubated for 1 h at RT with a secondary antibody conjugated with AlexaFluor 488. Subsequently, the EdU signal was developed using an EdU staining kit according to the manufacturer’s instructions. Finally, cells were stained with 40 μg/mL PI. Cell cycle specific analysis was carried out using flow cytometry combined with quantification by Kaluza 1.3, as described earlier [26,27].

### 4.6. Statistical Analysis

*p* values were calculated using t-test in Sigma Plot 11.0 and 14.0. The significance of differences between individual measurements is indicated by the * symbol representing *p* values: * *p* < 0.05, ** *p* < 0.01, *** *p* < 0.001; n.s. non-significant.

## Figures and Tables

**Figure 1 ijms-23-07506-f001:**
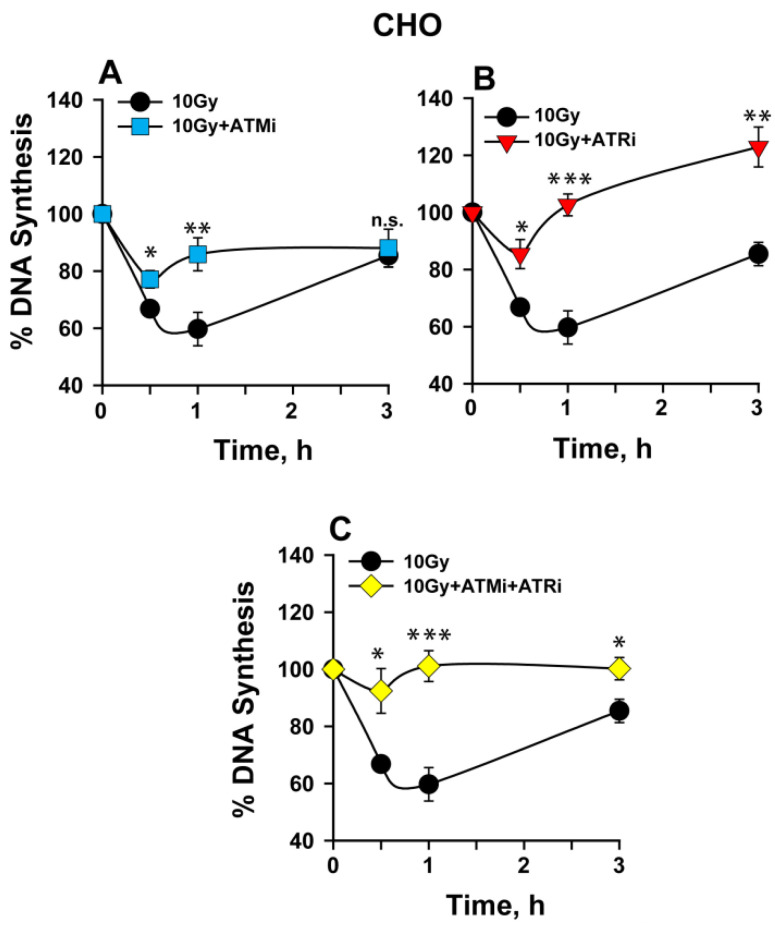
IR-induced intra-S-phase checkpoint in CHO cells. (**A**) Percentage of DNA synthesis over unirradiated controls measured after exposure to 10 Gy in the presence or absence of ATMi. (**B**) As in (**A**) for cells treated with ATRi. (**C**) As in (**A**) for cells treated with a combination of ATMi and ATRi. Error bars represent the standard deviation from 3 independent experiments. The significance of differences between individual measurements is indicated by * symbol: * *p* < 0.05, ** *p* < 0.01, *** *p* < 0.001; n.s. non-significant.

**Figure 2 ijms-23-07506-f002:**
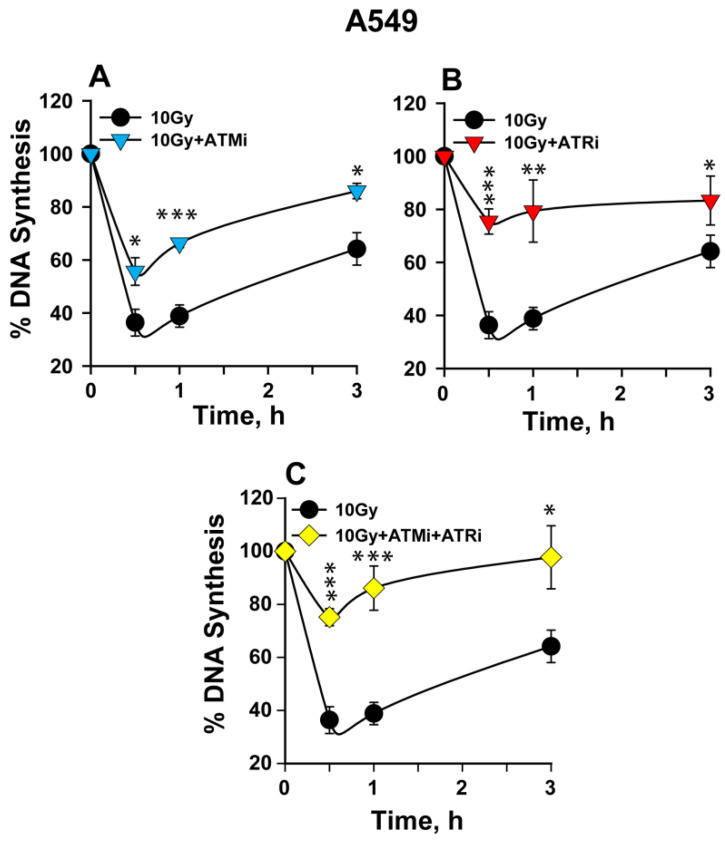
IR-induced intra-S-phase checkpoint in human A549 cells. (**A**) Percentage of DNA synthesis over unirradiated controls measured after exposure to 10 Gy of IR in the presence or absence of ATMi. (**B**) As in (**A**) for cells treated with ATRi. (**C**) As in (**A**) for cells treated with a combination of ATMi and ATRi. Error bars represent standard deviations from 3 to 4 independent experiments. The significance of differences between individual measurements is indicated by * symbol: * *p* < 0.05, ** *p* < 0.01, *** *p* < 0.001.

**Figure 3 ijms-23-07506-f003:**
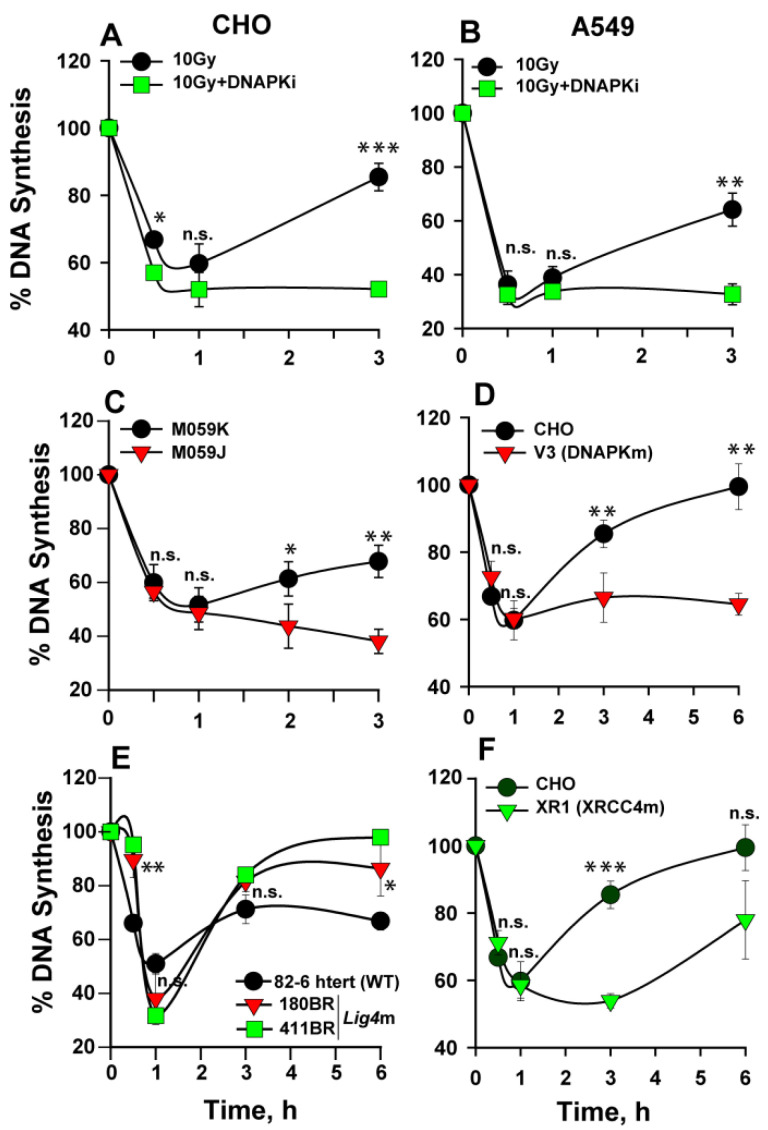
DNA-PK contributes to the recovery from the intra-S-phase checkpoint. (**A**) Percentage of DNA synthesis over unirradiated controls measured after exposure to 10 Gy of IR in the presence or absence of a DNA-PK inhibitor in CHO cells. (**B**) As in (**A**) for A549 cells. (**C**) As in (**A**) for M059K (wt) and M059J (DNA-PKcs mutant) cells. (**D**) As in (**A**) for CHO wt and the DNA-PK mutant, V3. (**E**) As in (**A**) for human fibroblast wt and Lig4 mutant cell lines. (**F**) As in (**A**) for CHO wt and the XRCC4 mutant, XR1. Error bars represent standard deviation from 3 independent experiments. The significance of differences between individual measurements is indicated by * symbol: * *p* < 0.05, ** *p* < 0.01, *** *p* < 0.001; n.s. non-significant. Results for 411BR cells are derived from a single experiment.

**Figure 4 ijms-23-07506-f004:**
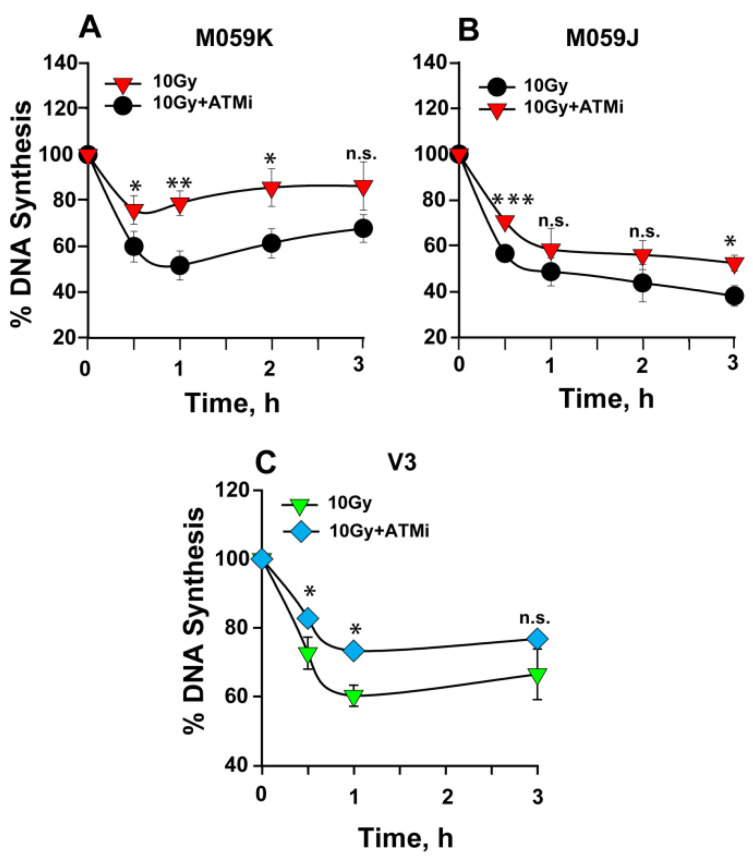
ATM has only a small contribution to the persistent intra-S-phase checkpoint in DNA-PKcs–deficient cells. (**A**) Percentage of DNA synthesis over non-irradiated controls measured after exposure to 10 Gy of IR in the presence or absence of ATMi in M059K cells. (**B**) As in (**A**) in M059J cells. (**C**) As in (**A**) in V3 cells. Error bars represent the standard deviation from 3 independent experiments. The significance of differences between individual measurements is indicated by * symbol: * *p* < 0.05, ** *p* < 0.01, *** *p* < 0.001; n.s. non-significant.

**Figure 5 ijms-23-07506-f005:**
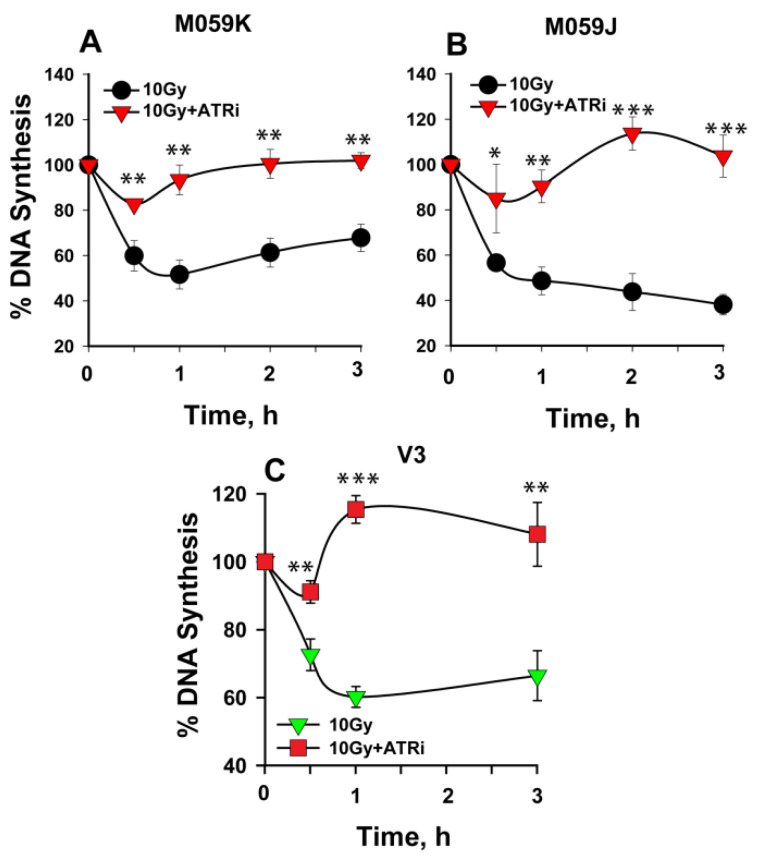
ATR is the key activator of the persistent intra-S-phase checkpoint in DNA-PKcs mutants. (**A**) Percentage of DNA synthesis over that of unirradiated controls measured after exposure to 10 Gy of IR in the presence or absence of ATRi in M059K cells. (**B**) As in (**A**) for M059J cells. (**C**) As in (**A**) for V3 cells. Error bars represent the standard deviation from 3 independent experiments. The significance of differences between individual measurements is indicated by * symbol: * *p* < 0.05, ** *p* < 0.01, *** *p* < 0.001.

**Figure 6 ijms-23-07506-f006:**
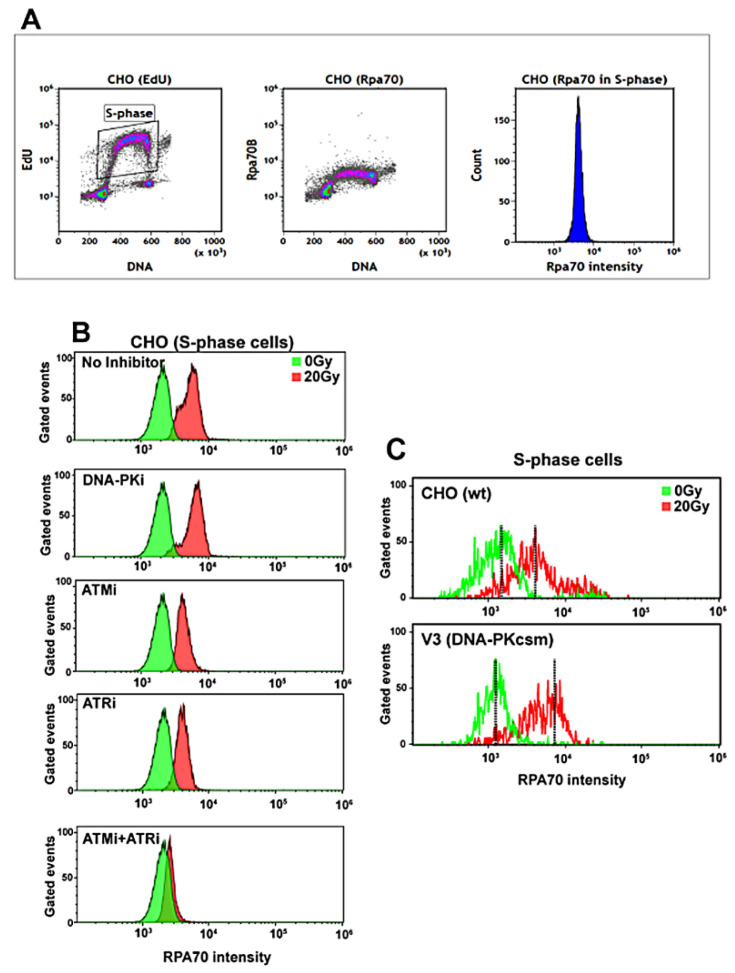
Degree of resection in S-phase cells correlates with intra-S-phase checkpoint activation: (**A**) Representative FACS histograms showing the S-phase specific gating scheme used to measure resection through analysis of RPA70 signal at 3 h post-IR, specifically in cells irradiated during S phase. (**B**) FACS overlays of CHO cells treated without or with the indicated PI3KK inhibitors. (**C**) FACS overlays of RPA70 signal in CHO and V3 cells exposed to the indicated doses of IR. Other details as in (**A**).

**Figure 7 ijms-23-07506-f007:**
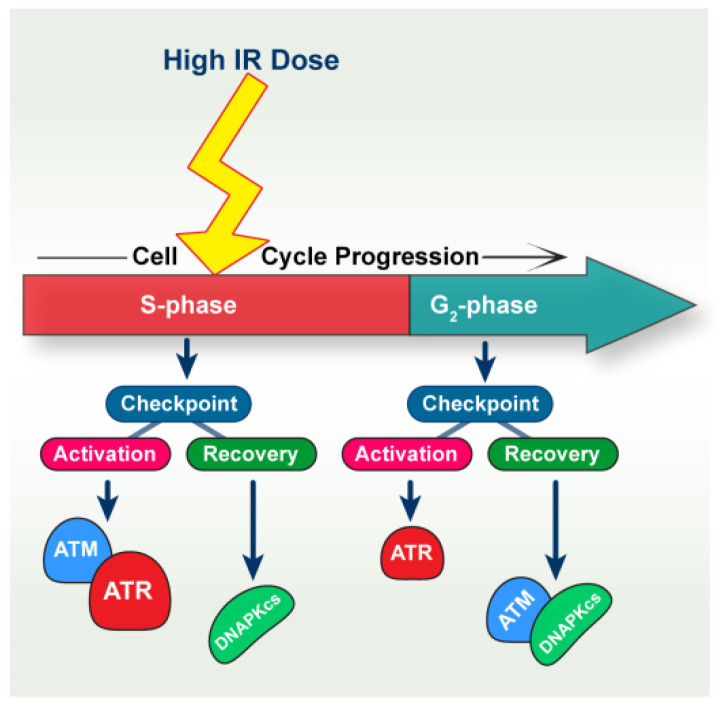
A model depicting the crosstalk between ATM, ATR, and DNA-PKcs in the regulation of checkpoint activation and recovery for S-phase cells exposed to high IR doses. The checkpoint is measured either in S phase (intra-S-phase checkpoint), as described in the present paper, or after progression of cells into G_2_ phase, as described earlier [27]. See discussion for further details.

## Data Availability

Not applicable.

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
