# Peer review of "ATR Contributes More Than ATM in Intra-S-Phase Checkpoint Activation after IR, and DNA-PKcs Facilitates Recovery: Evidence for Modular Integration of ATM/ATR/DNA-PKcs Functions"

_ijms, 2022, doi:10.3390/ijms23147506_

Round 1

Reviewer 1 Report

Soni et al. describe their findings on intra-S-phase checkpoint activation after IR upon perturbation of ATM, ATR, and DNA-PKcs. This work could be of interest to readers interested in the inter-S-phase checkpoint regulation in relation to applications such as cancer treatment.  However, the overall amount of primary data is lacking especially since there is no supplementary data to support the claims of the paper.  This should be addressed in order to improve the manuscript.

1. Primary data: the main issue of the paper is that it does not provide sufficient primary data. They perform tests in two or at most three cell lines for each condition (e.g., CHO and A549 for ATMi, ATRi tests). The authors need to provide more human cell lines (at least two more) to support their findings. 

2. Statistical test: the description for statistical test is lacking (only appears in methods section) for most figures. For Figure 3, no statistical test was performed. For all other figures, only a single point (at 1 hr) was usually tested for statistical differences. They should extend statistical tests to all data points. In Figure 3E, there is no replicate and there should be at least 3 replicates in order to present biological data.

3. Overall picture: the authors need to provide a summary figure to present their findings (e.g., as Figure 7). Otherwise, it is difficult to grasp what their finding was.  

Overall, the authors need to perform a number of experiments to support their claims.

Author Response

IJMS-1772430

Response to Reviewers' Comments

We are thankful to both the reviewers for the overall positive evaluation of our work. Their comments and criticisms helped us improve the manuscript. Below we provide a point-by-point response to the Reviewers’ comments. We submit a “clean” copy of the manuscript and include a “marked copy” generated using the “compare” function of “Word” as “Supplementary Material” to facilitate the visualization of changes made.

Our specific responses are as follows:

Reviewer #1

Soni et al. describe their findings on intra-S-phase checkpoint activation after IR upon perturbation of ATM, ATR, and DNA-PKcs. This work could be of interest to readers interested in the inter-S-phase checkpoint regulation in relation to applications such as cancer treatment.  However, the overall amount of primary data is lacking especially since there is no supplementary data to support the claims of the paper.  This should be addressed in order to improve the manuscript.

We thank the Reviewer for the positive assessment of our paper. In the revised version of the paper, we have included the results of additional experiments (also requested by reviewer 2) as main and supplementary figures to further strengthen the statistical weight, the conclusions, and interpretations of our paper, and to increase its footprint!

  1. Primary data: the main issue of the paper is that it does not provide sufficient primary data. They perform tests in two or at most three cell lines for each condition (e.g., CHO and A549 for ATMi, ATRi tests). The authors need to provide more human cell lines (at least two more) to support their findings.

As per the reviewer’s suggestion, we included data from additional human cell lines in the revised version. The human cell lines included now in the paper are: A549, M059K, M059J, HCT116 wt, LIG4-/-, DNA-PK-/-, 82-6hTert, 180BRIM, and 411BR. We also include results with the Chinese Hamster cell lines, CHO, V3, and XR1. We measured the impact of ATMi and ATRi on IR-induced intra-S-phase checkpoint in six cell lines. The results in aggregate support our conclusions and interpretations and helped us to develop the model we present.

  1. Statistical test: the description for statistical test is lacking (only appears in methods section) for most figures. For Figure 3, no statistical test was performed. For all other figures, only a single point (at 1 hr) was usually tested for statistical differences. They should extend statistical tests to all data points. In Figure 3E, there is no replicate and there should be at least 3 replicates in order to present biological data.

We thank the Reviewer for raising this rather important point. Details of statistical tests are now added in each figure legend. In addition, statistical analysis is performed for all data points, including Figure 3. The data presented in Figure 3E of the revised version is obtained from three experiments for 82-6 hTert and 180BR cells. We were unable to repeat the experiment with 411BR LIG4 mutant owing to the fact that these cells are not immortalized, and still available stocks grow suboptimally at the moment. In addition, we included data from XR1 cells (Figure 3F) which better correlate to CHO cells (please also see our response to Reviewer 2).

  1. Overall picture: the authors need to provide a summary figure to present their findings (e.g., as Figure 7). Otherwise, it is difficult to grasp what their finding was.

This is an excellent suggestion. In the revised version, we have included a summary figure (Figure 7) of our observations and conclusions.

Reviewer #2

This manuscript addresses an important point about the functions of ATM and ATR kinases. Based on mutants, ATM has been advertised as the main or only kinase involved in the intra-S checkpoint, possibly because ATR loss of function mutants are not viable. The authors show convincing evidence that ATR has an important function here.

We thank the reviewer for the positive assessment of our paper and for recognizing our findings and their significance in placing ATR at the forefront of the regulation of the IR-induced intra-S phase checkpoint in mammalian cells.

The function of DNA-PK is a bit less clear. The authors should be careful not to overstate their findings. Although all observations are in the same direction, the effects are not very strong. Furthermore, their models are suboptimal: DNA-PK inhibitors may prevent repair of DNA breaks, confounding interpretation of results, the MO59J model is not only DNA-PKcs deficient, but also ATM low, and the CHO models cannot easily be compared to human fibroblasts (180BR and 411BR). For these CHO models, I would at least have expected the comparison to XR1 (XRCC4 deficient) and Ku80-deficient cells. Also repaired version of the V3 cells would be available (expressing the human DNA-PKcs) could be used here. If the authors want to keep in a strong statement about DNA-PK function in recovery from the intra-S checkpoint, some of these experiments should be included to strengthen the argument. Also, a few more time points (e.g. also 2 and 4 hours) could help discriminate between a delay in recovery and no recovery at all.

We thank the reviewer for these relevant suggestions regarding DNA-PKcs function in the intra-S-phase checkpoint. As per reviewer’s suggestion, we have included in the revised manuscript the data for XR1 cells to directly compare with CHO models. We also added results with human, HCT116 LIG4-/- and DNA-PKcs-/- cells (Figure S1). Although HCT116 LIG4-/-, and XR1 cells, in contrast to other LIG4 deficient cells, exhibited checkpoint potentiation when compared to WT cells, they show clear recovery from the intra-S-phase checkpoint (Figures 3F, and Figure S1) when compared to DNA-PKcs deficient cells (Figures 3D, and Figure S1). In addition, we have included results on ATMi’s and ATRi’s impact on the intra-S-phase checkpoint in XR1 cells as supplementary Figure 2. In these new experiments, extended times have been applied (up to 6 hours) as also suggested by the reviewer.

We would also like to mention the findings of a previous publication from our group (Guan J. et al., 2000, Nucleic Acids Research, Reference 24), which also implicate DNA-PKcs in the recovery from the intra-S-phase checkpoint. This offer additional and independent evidence for the conclusions of the present paper regarding the function of DNA-PKcs. In the present paper, our main goal was to investigate the roles of ATM and/or ATR in this sustained checkpoint in cells with ablated DNA-PKcs function.

Minor point:

Page 2 states that ‘DNA damage signaling’ is abbreviated as DDR. The normal meaning of DDR is DNA damage response, which is broader than signaling. The authors should therefore not use this abbreviation for only the signaling component of the response.

We agree and have modified the corresponding passage according to the reviewer’s suggestion.

Reviewer 2 Report

This manuscript addresses an important point about the functions of ATM and ATR kinases. Based on mutants, ATM has been advertised as the main or only kinase involved in the intra-S checkpoint, possibly because ATR loss of function mutants are not viable. The authors show convincing evidence that ATR has an important function here.

The function of DNA-PK is a bit less clear. The authors should be careful not to overstate their findings. Although all observations are in the same direction, the effects are not very strong. Furthermore, their models are suboptimal: DNA-PK inhibitors may prevent repair of DNA breaks, confounding interpretation of results, the MO59J model is not only DNA-PKcs deficient, but also ATM low, and the CHO models cannot easily be compared to human fibroblasts (180BR and 411BR). For these CHO models, I would at least have expected the comparison to XR1 (XRCC4 deficient) and Ku80-deficient cells. Also repaired version of the V3 cells would be available (expressing the human DNA-PKcs) could be used here. If the authors want to keep in a strong statement about DNA-PK function in recovery from the intra-S checkpoint, some of these experiments should be included to strengthen the argument. Also a few more time points (e.g. also 2 and 4 hours) could help discriminate between a delay in recovery and no recovery at all.

Minor point:

Page 2 states that ‘DNA damage signaling’ is abbreviated as DDR. The normal meaning of DDR is DNA damage response, which is broader than signaling. The authors should therefore not use this abbreviation for only the signaling component of the response.

Author Response

IJMS-1772430

Response to Reviewers' Comments

We are thankful to both the reviewers for the overall positive evaluation of our work. Their comments and criticisms helped us improve the manuscript. Below we provide a point-by-point response to the Reviewers’ comments. We submit a “clean” copy of the manuscript and include a “marked copy” generated using the “compare” function of “Word” as “Supplementary Material” to facilitate the visualization of changes made.

Our specific responses are as follows:

Reviewer #1

Soni et al. describe their findings on intra-S-phase checkpoint activation after IR upon perturbation of ATM, ATR, and DNA-PKcs. This work could be of interest to readers interested in the inter-S-phase checkpoint regulation in relation to applications such as cancer treatment.  However, the overall amount of primary data is lacking especially since there is no supplementary data to support the claims of the paper.  This should be addressed in order to improve the manuscript.

We thank the Reviewer for the positive assessment of our paper. In the revised version of the paper, we have included the results of additional experiments (also requested by reviewer 2) as main and supplementary figures to further strengthen the statistical weight, the conclusions, and interpretations of our paper, and to increase its footprint!

  1. Primary data: the main issue of the paper is that it does not provide sufficient primary data. They perform tests in two or at most three cell lines for each condition (e.g., CHO and A549 for ATMi, ATRi tests). The authors need to provide more human cell lines (at least two more) to support their findings.

As per the reviewer’s suggestion, we included data from additional human cell lines in the revised version. The human cell lines included now in the paper are: A549, M059K, M059J, HCT116 wt, LIG4-/-, DNA-PK-/-, 82-6hTert, 180BRIM, and 411BR. We also include results with the Chinese Hamster cell lines, CHO, V3, and XR1. We measured the impact of ATMi and ATRi on IR-induced intra-S-phase checkpoint in six cell lines. The results in aggregate support our conclusions and interpretations and helped us to develop the model we present.

  1. Statistical test: the description for statistical test is lacking (only appears in methods section) for most figures. For Figure 3, no statistical test was performed. For all other figures, only a single point (at 1 hr) was usually tested for statistical differences. They should extend statistical tests to all data points. In Figure 3E, there is no replicate and there should be at least 3 replicates in order to present biological data.

We thank the Reviewer for raising this rather important point. Details of statistical tests are now added in each figure legend. In addition, statistical analysis is performed for all data points, including Figure 3. The data presented in Figure 3E of the revised version is obtained from three experiments for 82-6 hTert and 180BR cells. We were unable to repeat the experiment with 411BR LIG4 mutant owing to the fact that these cells are not immortalized, and still available stocks grow suboptimally at the moment. In addition, we included data from XR1 cells (Figure 3F) which better correlate to CHO cells (please also see our response to Reviewer 2).

  1. Overall picture: the authors need to provide a summary figure to present their findings (e.g., as Figure 7). Otherwise, it is difficult to grasp what their finding was.

This is an excellent suggestion. In the revised version, we have included a summary figure (Figure 7) of our observations and conclusions.

Reviewer #2

This manuscript addresses an important point about the functions of ATM and ATR kinases. Based on mutants, ATM has been advertised as the main or only kinase involved in the intra-S checkpoint, possibly because ATR loss of function mutants are not viable. The authors show convincing evidence that ATR has an important function here.

We thank the reviewer for the positive assessment of our paper and for recognizing our findings and their significance in placing ATR at the forefront of the regulation of the IR-induced intra-S phase checkpoint in mammalian cells.

The function of DNA-PK is a bit less clear. The authors should be careful not to overstate their findings. Although all observations are in the same direction, the effects are not very strong. Furthermore, their models are suboptimal: DNA-PK inhibitors may prevent repair of DNA breaks, confounding interpretation of results, the MO59J model is not only DNA-PKcs deficient, but also ATM low, and the CHO models cannot easily be compared to human fibroblasts (180BR and 411BR). For these CHO models, I would at least have expected the comparison to XR1 (XRCC4 deficient) and Ku80-deficient cells. Also repaired version of the V3 cells would be available (expressing the human DNA-PKcs) could be used here. If the authors want to keep in a strong statement about DNA-PK function in recovery from the intra-S checkpoint, some of these experiments should be included to strengthen the argument. Also a few more time points (e.g. also 2 and 4 hours) could help discriminate between a delay in recovery and no recovery at all.

We thank the reviewer for these relevant suggestions regarding DNA-PKcs function in the intra-S-phase checkpoint. As per the reviewer’s suggestion, we have included in the revised manuscript the data for XR1 cells to directly compare with CHO models. We also added results with human, HCT116 LIG4-/- and DNA-PKcs-/- cells (Figure S1). Although HCT116 LIG4-/-, and XR1 cells, in contrast to other LIG4 deficient cells, exhibited checkpoint potentiation when compared to WT cells, they show clear recovery from the intra-S-phase checkpoint (Figures 3F, and Figure S1) when compared to DNA-PKcs deficient cells (Figures 3D, and Figure S1). In addition, we have included results on ATMi’s and ATRi’s impact on the intra-S-phase checkpoint in XR1 cells as supplementary Figure 2. In these new experiments, extended times have been applied (up to 6 hours) as also suggested by the reviewer.

We would also like to mention the findings of a previous publication from our group (Guan J. et al., 2000, Nucleic Acids Research, Reference 24), which also implicate DNA-PKcs in the recovery from the intra-S-phase checkpoint. This offer additional and independent evidence for the conclusions of the present paper regarding the function of DNA-PKcs. In the present paper, our main goal was to investigate the roles of ATM and/or ATR in this sustained checkpoint in cells with ablated DNA-PKcs function.

Minor point:

Page 2 states that ‘DNA damage signaling’ is abbreviated as DDR. The normal meaning of DDR is DNA damage response, which is broader than signaling. The authors should therefore not use this abbreviation for only the signaling component of the response.

We agree and have modified the corresponding passage according to the reviewer’s suggestion.